# Design and Synthesis of Benzene Homologues Tethered with 1,2,4-Triazole and 1,3,4-Thiadiazole Motifs Revealing Dual MCF-7/HepG2 Cytotoxic Activity with Prominent Selectivity via Histone Demethylase LSD1 Inhibitory Effect

**DOI:** 10.3390/ijms23158796

**Published:** 2022-08-08

**Authors:** Mosa Alsehli, Ateyatallah Aljuhani, Saleh K. Ihmaid, Shahenda M. El-Messery, Dina I. A. Othman, Abdel-Aziz A. A. El-Sayed, Hany E. A. Ahmed, Nadjet Rezki, Mohamed R. Aouad

**Affiliations:** 1Chemistry Department, College of Sciences, Taibah University, Al-Madinah Al-Munawarah 41477, Saudi Arabia; 2Pharmacognosy and Pharmaceutical Chemistry Department, College of Pharmacy, Taibah University, Al-Madinah Al-Munawarah 42353, Saudi Arabia; 3Pharmaceutical Chemistry Department, Faculty of Pharmacy, Jadara University, Irbid 21110, Jordan; 4Department of Pharmaceutical Organic Chemistry, Faculty of Pharmacy, Mansoura University, Mansoura 35516, Egypt; 5Biology Department, Faculty of Science, Islamic University of Madinah, Al-Madinah Al-Munawarah 42351, Saudi Arabia; 6Zoology Department, Faculty of Science, Zagazig University, Zagazig 44519, Egypt; 7Pharmaceutical Organic Chemistry Department, Faculty of Pharmacy, Al-Azhar University, Cairo 35511, Egypt

**Keywords:** 1,2,4-triazoles, 1,3,4-thiadiazoles, LSD1, melanoma, tris-substituted analogues, anticancer activity, apoptosis study

## Abstract

In this study, an efficient multistep synthesis of novel aromatic tricyclic hybrids incorporating different biological active moieties, such as 1,3,4-thiadiazole and 1,2,4-triazole, was reported. These target scaffolds are characterized by having terminal lipophilic or hydrophilic parts, and their structures are confirmed by different spectroscopic methods. Further, the cytotoxic activities of the newly synthesized compounds were evaluated using in vitro MTT cytotoxicity screening assay against three different cell lines, including HepG-2, MCF-7, and HCT-116, compared with the reference drug Taxol. The results showed variable performance against cancer cell lines, exhibiting MCF-7 and HepG-2 selectivities by active analogs. Among these derivatives, 1,2,4-triazoles **11** and **13** and 1,3,4-thiadiazole **18** were found to be the most potent compounds against MCF-7 and HepG-2 cancer cells. Moreover, structure–activity relationship (SAR) studies led to the identification of some potent LSD1 inhibitors. The tested compounds showed good LSD1 inhibitory activities, with an IC_50_ range of 0.04–1.5 μM. Compounds **27**, **23**, and **22** were found to be the most active analogs with IC_50_ values of 0.046, 0.065, and 0.074 μM, respectively. In addition, they exhibited prominent selectivity against a MAO target with apparent cancer cell apoptosis, resulting in DNA fragmentation. This research provides some new aromatic-centered 1,2,4-triazole-3-thione and 1,3,4-thiadiazole analogs as highly effective anticancer agents with good LSD1 target selectivity.

## 1. Introduction

Cancer is a serious medical problem and one of the leading causes of mortality in the globe [1,2]. Although survival rates for some cancers have improved in recent decades as a result of early detection and the introduction of novel medicines, effective therapeutic options for the successful treatment of advanced tumors remain inadequate [3]. Clinical data suggest that a lack of early detection is responsible for around 80% of cancer-related fatalities [4,5]. Traditional cancer treatments, such as radiation and chemotherapy, are no longer effective because they kill normal and malignant cells at similar rates, resulting in substantial damage and very little to moderate therapeutic gains [6,7]. Nowadays, a variety of organic molecules are designed for this purpose using different types of molecular hybridization protocols to attach a structural feature or modify functional groups into old or new chemical moieties [8]. Nitrogen-based heterocycles are particularly important in the development of anticancer drugs since they appear in nearly three-quarters of the heterocyclic anticancer medicines authorized by the FDA [9]. Specifically, benzoheterocyclic and triazole compounds have a wide range of biological applications and have been highlighted as one of the most promising drug scaffolds in pharmacology [10]. Five-membered nitrogen heterocyclic systems are among the most valuable nitrogen containing heterocycles having the ability to induce cell death in various cancer cell lines [11,12]. Because of their numerous biological actions, such as anti-inflammatory and antiproliferative properties, 1,2,4-triazole derivatives have become medicinal medicines in recent decades [13]. This nucleus provides appropriate water solubility and flexibility and interacts with certain enzymes involved in cancer development due to its ion chelating properties and the ability to form hydrogen bonding through its –NH– group [14]. Furthermore, having a triazole moiety increases liver microsomal stability and antitumor activity [15,16]. On the other hand, 1,3,4-thiadiazole derivatives have attracted considerable interest owing to their wide spectrum of biological activity, including antimicrobial, antituberculosis, anticonvulsant, anti-inflammatory, anticancer, and antiulcer properties [17,18,19,20,21,22,23,24,25,26,27,28,29]. The lysine-specific histone demethylase 1A (LSD1, also known as KDM1A) removes methyl marks on histone H3 lysine 4 (H3K4) in a flavin adenine dinucleotide (FAD)-dependent way among epigenetic targets [30]. LSD1 plays a variety of functions in physiological and pathological processes, including cancer [31], infections [32], and immunological regulation [33]. Because LSD1 is overexpressed in a number of human cancers (acute myeloid leukemia, neuroblastoma, retinoblastoma, prostate cancer, breast cancer, lung cancer, and bladder cancer and melanoma types) [32,33,34], the protein has emerged as an important target for the development of specific inhibitors as a new class of antitumor agents (Figure 1) [35,36]. It has been reported that LSD1 has the ability to suppress melanoma through an increase in repetitive element expression, including ERVs, and decreases the expression of RNA-induced silencing complex (RISC) components. Significantly, this leads to dsRNA stress and activation of type 1 interferon, which stimulates anti-tumor T cell immunity and restrains tumor growth [33]. Other mechanism for LSD1 in melanoma is geometric regulation of the histone state, which succeeds in direct melanoma reprogramming and hence affects growth [37].

As part of our research work toward the synthesis of heterocycles with broad biological activities [21,22,23,24,25,26,27,28,29,38], this work aims to synthesize a library of some new aromatic-centered homologues tethered with two different active motifs: 1,2,4-triazole-3-thione and 1,3,4-thiadiazole. The design is based upon the chemical study of LSD1 inhibitors [39,40,41,42,43] in Figure 1 through the molecular hybridization of a central aromatic system with pyridine, pyrimidine, and imidazole rings in compounds **1**–**4**. Compounds **5**–**9** are embedded with five-membered aromatic systems, triazole and thiadiazole, which show good affinity and potency against LSD1. However, gathering all SAR analysis of these leads as reported LSD1 inhibitors motivated us to introduce our target scaffold composed of an aromatic ring with three homogenous trisubstitutions with corresponding triazole or thiadiazole heterocycles. The molecular docking of target compounds revealed the good placement within the LSD1 pocket, and their anticancer activity against different cancers was investigated. Moreover, the inhibitory activity evaluation against LSD1 has been investigated as a proposed mechanism of their anticancer activity. These analogues offer a lot of therapeutic promise for cancer treatment, as well as other epigenetically driven diseases.

## 2. Results and Discussion

### 2.1. Chemistry 

The synthetic pathways adopted in this study are depicted in Figure 1 and Figure 2. The targeted triheterocyclic molecules were multistep-synthesized starting from benzene-1,3,5-tricarbohydrazide **11** as precursor. Compound **11** was successfully prepared in 94% yield through treatment of triethyl benzene-1,3,5-tricarboxylate **10** (reported compound [44]) with hydrazine hydrate for 4 h in refluxing ethanol (Figure 1). The resulting acid hydrazide **11** was confirmed clearly through the appearance of a new absorption band near 3250–3390 cm^−1^ belonging to the hydrazide side chains (–NHNH_2_) in its IR spectrum. Additionally, the ^1^H and ^13^C NMR spectra of our hydrazide **11** were characterized by the absence of the aliphatic ethoxy protons and carbons of the corresponding tris-ester **10**. New signals were also observed at 9.86 and 4.59 ppm assignable to the hydrazide NH and NH_2_ protons, respectively. The three aromatic protons resonated as one singlet at 8.33 ppm. In addition, the three amidic carbonyl (C=O) carbons were recorded at 165.47 ppm in the ^13^C NMR spectrum.

The condensation of the tris-acid hydrazide **11** with a variety of alkyl/aryl isothiocyanate under refluxing ethanol afforded the corresponding tris-acid thiosemicarbazides **12**–**17** (87–90% yields) (Figure 1). Compounds **12**–**17** were fully characterized by IR and NMR analysis. All IR spectra confirm the proposed structures by the appearance of new bands around 1280–1300 cm^−1^ attributed to the thiocarbonyl group (C=S). Their ^1^H NMR spectra showed new diagnostic signals at 8.60–10.95 ppm assigned to the thiosemicarbazide NH protons (CONH and NHCSNH) with no observed NH_2_ protons. Particularly, the ^1^H-NMR spectrum of compound **12** revealed the presence of a singlet at 3.39 ppm related to the three equivalent methyl groups. The signal belonging to the same methyl carbons were observed at 31.44 ppm in its ^13^C-NMR spectrum. In addition, the three thiocarbonyl carbons (C=S) resonated at 182.70 ppm, while the aromatic carbons were recorded at 130.84 and 133.34 ppm, see the Appendix A.

Then, dehydrative intramolecular cyclization was carried out to get the targeted tris-1,2,4-triazole-3-thiones **18**–**23**. Thermal intramolecular cyclodehydration of the acid thiosemicarbazides **12**–**17** in basic media (10% NaOH) gave the corresponding 1,2,4-triazole-3-thiones **18**–**23** in 85–88% yields (Figure 2). The structures of the resulting triazoles **18**–**23** were elucidated from their spectral analysis. Their IR spectra showed common characteristic absorption peaks at 3280–3340 cm^−1^ assigned to (NH), 1615–1630 cm^−1^ (C=N), and 1290–1310 cm^−1^ (C=S), which supported the formation of triazole rings. It is worth mentioning that there is a possible tautomerism of the resulting tris-1,2,4-triazoles **18**–**23,** which could exist in two tautomeric forms: a thiol form and a thione form. On the basis of the obtained NMR spectra, it was recognized that the thione form is the most stable form in which these derivatives exist [45]. The ^1^H-NMR spectra of compounds **18**–**23** revealed the absence of the characteristic NH protons of their thiosemicarbazide precursors **12**–**17** and the presence of distinguishable triazolic NH singlets around 12.90–14.25 ppm, which is clear evidence for the success of the intramolecular cyclization reaction forming the three 1,2,4-triazole cores in the thione form. In addition, the signals belonging to the C=S carbons were recorded at 166.12–190.51 ppm supporting the prevalence of the thione isomers.

On the other hand, the action of sulfuric acid on the thiosemicarbazide intermediates **12**–**17** at 0 °C (Figure 2) furnished the desired 2-amino-1,3,4-thiadiazoles **24**–**28** in good yields (83–85%).

The structures of the newly designed tris-thiadiazoles **24**–**28** were deduced from their spectroscopic results, which disclosed the appearance of characteristic absorption bands near 1600–1620 cm^−1^ assigned to the C=N group and the disappearance of the absorption bands of the thiocarbonyl (C=S) and carbonyl (C=O) groups in their IR spectra. The ^1^H and ^13^C-NMR spectra also confirmed the absence of the signals attributed to the characteristic protons and carbons of their corresponding thiosemicarbazide precursors **3**–**8** (i.e., –CONH and –NHCSNH–) around 8.60–10.95 ppm and 164.87–190.76 ppm. In addition, their ^1^H-NMR spectra revealed the presence of the diagnostic exocyclic NH protons at position 2 of the 1,3,4-thiadiazole ring at 6.99–7.63 ppm. All these results confirmed the acid-catalyzed intramolecular ring closure of thiosemicarbazide intermediates **12**–**17,** which afforded the desired tris-1,3,4-thiadiazoles **24**–**28**. All compounds were precipitated and recrystallized, affording yellowish to white colors and crystalline powders.

### 2.2. Biological Screening

#### 2.2.1. In Vitro Antitumor Activity

All the newly synthesized compounds were evaluated for their in vitro anticancer effect via the standard MTT method against a panel of three different human cancer cell lines: breast (MCF-7), colon (HCT-116), and liver (HepG_2_) [22,24,26]. As shown in Table 1, the obtained results revealed that the tested compounds **19**, **21**, **22**, **24**, **27**, and **28** exhibited very strong potency against a tested breast cancer cell line with an IC_50_ ranging from 1.52 to 9.60 μM, in which compound **27** exhibited the most active candidate with an IC_50_ of 1.52 μM compared with the reference drug Taxol (IC_50_ = 7.80 μM). Moreover, compounds **20** and **26** showed strong activity against breast cell lines with an IC_50_ of 13.7–17.1 μM, respectively. In addition, compounds **21**, **22**, and **23** exhibited very strong potency against a tested colon cancer cell line with an IC_50_ ranging from 2.01 to 5.27 μM, where compound **23** exhibited the most active candidate with an IC_50_ of 2.01 μM compared with Taxol’s IC_50_ at 7.96 μM. Furthermore, compounds **20**, **24**, **26**,and **27** showed strong activity against the same cell line with an IC_50_ ranging from 10.1–18.1 μM. For a liver cancer cell line, the obtained results revealed that the tested compounds **20**, **21**, **22**, **23**, and **24** exhibited very strong potency with an IC_50_ ranging from 2.36 to 8.23 μM, in which compound **21** exhibited the most active candidate with an IC_50_ of 2.36 μM compared with Taxol’s IC_50_ = 4.44 μM. Moreover, compound **25** showed strong activity against the same cell line with IC_50_ =13.0 μM. Moreover, compound **18** showed moderate activity against all the tested cell lines with IC_50_ ranging from 23.9 to 38.6 μM.

On the other hand, compounds **28** and **26** showed the lowest potency and least inhibitory activity against HCT116 with IC_50_ = 110.0 μM and HepG2 cell lines with IC_50_ = 95.7 μM. In order to investigate whether the newly synthesized compounds exhibited selective activity against cancer and normal cells, compounds **22** (nonselective)and **27** (MCF-7 selective) were selected for screening their cytotoxic activity on a normal human lung fibroblast(WI38) cell line to determine its therapeutic safety. As shown in Table 1, they displayed low cytotoxicity against a normal (WI38) cell line with IC_50_ =16.0 and 85 µM compared with Taxol with IC_50_ = 11.8 µM. 

#### 2.2.2. LSD_1_ Demethylase Inhibition Assay

LSD1 demethylase has also been linked to various pathological processes, such as the development of cancer [46,47]. LSD1 inhibition may result in H3K4 remethylation and silencing of H3K4-enriched active genes. As shown in Table 2, compounds **18**–**28** were selected for LSD_1_ inhibition assay against GSK-LSD_1_ as a reference drug control [48]. The results demonstrated that most of the tested compounds **22**, **23**, and **27** exhibited very strong LSD_1_ inhibitory activity with an IC_50_ value of 0.074, 0.065, and 0.046 µM, respectively. Meanwhile, the compounds **19**, **20**, **21**, **24**, **25**, **26**, and **28** showed moderate LSD_1_ inhibitory activity with an IC_50_ ranging from 0.13 to 0.31 μM. Compound **18** showed weak LSD_1_ inhibitory activity with an IC_50_ value of 1.51 µM.

#### 2.2.3. Cell Cycle Analysis 

The cell cycle is a series of growth and development processes that culminate in DNA replication and cell division [49]. G1 phase, S phase (synthesis), G2 phase, and M phase are the four different phases. We chose our promising chemical 27 based on the results of the MTT assay (Table 1) to further examine its effect on the apoptotic process and investigate its ability to play an active role in cell cycle progression in MCF-7 cells. MCF-7 cells were treated for 24 h with compound 27 at a concentration of 2.0 M, and flow cytometry was performed in the presence of annexin V-FITC/PI. Furthermore, when compared with the control, the distribution of cells in the G1 and S phases was significantly reduced in all evaluated cells (Figure 2). Cells treated with compound 27 had higher DNA content at the G2-M phase (35.59% versus 7.48% for the control) and the Pre-G1 phase (37.11% versus 1.36% for the control), revealing cell growth arrest at these two phases corresponding to the sum of annexin V-positive/PI-negative staining and double-positive staining cells (Table 3) (early and late apoptosis, respectively).

#### 2.2.4. Apoptosis Detection Using Annexin V-Fluorescein Isothiocyanate (FITC)/PI Dual Staining Assay

This test used the breast MCF-7 cell line to assess the percentage of apoptosis caused by compound **27**, in which MCF-7 cells were treated with compound **27** at 2.0 µM concentration for 24 h to measure the percentage of apoptosis induced by compound **27**. Compound **27** was found to cause early apoptosis (5.33%) in MCF-7 cells and enhance late apoptosis (20.53%) by 114-fold when compared with untreated cells. As a result, it is possible to deduce that compound **27** causes apoptosis at a rate of 37.11% (Table 4 and Figure 3 and Figure 4). These findings are in line with statistics derived from DNA content.

Figure 3 also exhibited the findings of fluorescence-activated cell sorting analysis performed after MCF-7 cells were treated with our compound **27**. Each sample’s histogram was separated into four quadrants to show viable (lower left quadrant), necrotic (higher left quadrant), apoptotic (lower right quadrant), and necrotic/late apoptotic (upper right quadrant) cells [22,38]. The Annexin V Apoptosis Detection Kit is based on the finding that apoptotic cells can transport phosphatidylserine (PS) from the plasma membrane’s inner surface to the cell surface. As a result, the presence of PS on the cell surface is regarded as an important technique for detecting apoptotic cells. PS can be easily detected by labeling with a fluorescent compound of annexin V, a protein with a high affinity for PS, and then analyzing the results by flow cytometry. When annexin V and PI staining are performed simultaneously, the kit can distinguish between apoptosis and necrosis. Only when the plasma membrane is completely damaged may PI enter the cell. This allows early apoptotic cells (negative for PI but positive for PS) to be distinguished from late apoptotic and necrotic cells (positive for both PI and PS). As shown in the cell cycle study of MCF-7, our tested compound **27** had a pre-G1 peak, indicating apoptosis. Cells were tested using annexin V and PI double-staining flow cytometry, which detect both PS residues and DNA, respectively, to validate the potential to trigger apoptosis. They were then cultured for 24 h before being examined. The tested compound was able to produce considerable levels of cell death in a dose-dependent manner, according to analyses of early and late apoptosis (Table 4, Figure 2 and Figure 3). MCF-7 cells treated with compound **27** had an increased number of early and late apoptotic cells when compared with the control. Compound **27** treatment of MCF-7 cells for 24 h resulted in 37.11% apoptotic cells compared with 1.36% apoptotic cells in the untreated control, as shown in Figure 4. In cancer therapy, the most effective and safe anticancer agents often disrupt the equilibrium between cell growth and apoptosis, causing cells to undergo apoptotic induction [50]. These findings demonstrate that compound **27** inhibits cell development by inducing cell apoptosis.

#### 2.2.5. DNA Fragmentation

Apoptosis is characterized by DNA fragmentation, which occurs in all stages of the process [51]. The percentage of fragmented DNA released into the cytoplasm from apoptotic nuclei was determined using a diphenylamine test. In the HepG2 and MCF-7 cells treated with selective compounds **20** and **27**, the relative quantity of DNA fragments is displayed in Figure 5. In all cells treated with these compounds **20** and 27 (approximately 28.5 ± 1.5% and 23 ± 1.2%, respectively), the percentage of DNA fragmentation rose significantly as compared with the untreated control (about 4.7 ± 0.2%), with no significant differences found between tested cells. These findings corroborated the apoptotic findings, indicating that the treatment of HepG2 and MCF-7 cells with compounds **20** and **27** caused cell membrane breakdown, followed by chromosomal DNA fragmentation.

#### 2.2.6. Target MAO Selectivity Analysis

Here, the potent selective compound **27** was screened for measuring target selectivity against high structural similar membrane-bound human *h*MAO-A and *h*MAO-B at 10 µM in the presence of tyramine at 2 × KM (0.8 mM for MAO-A and 0.32 mM for MAO-B) (Table 5). The data of the inhibitory assay revealed the molecular selectivity of **27** derivative for LSD_1_ over the MAO isoforms compared with the selective reference MAO drugs, as shown Table 5.

#### 2.2.7. Molecular Modeling Study

To gain insights into the mechanism of LSD_1_ inhibition and to elucidate the binding mode, in a comparative molecular modeling study, active derivatives **22** and **23** and **27** against compound **18**, which represents the least LSD1 inhibitory activity, were further subjected to a molecular modeling study to demonstrate the binding mode and the type of interaction at the molecular level. To date, there are a number of X-ray crystal structures of LSD_1_ in a complex with small-molecule and/or peptide inhibitors deposited in the RCSB protein database [53]. The crystal structure (PDB code: 5YJB) cocrystallized with 4-[5-(piperidin-4-ylmethoxy)-2-(p-tolyl)pyridin-3-yl]benzonitrile, which bears a 4-piperidinylmethoxy group, a 4-methylphenyl group, and a 4-cyanophenyl group on a pyridine ring, showing that its cyano group forms a hydrogen bond with Lys661, which is a critical residue in the lysine demethylation reaction located deep in the catalytic center of LSD_1_ [54,55].

It was recently reported to be a reversible LSD_1_ inhibitor with a Ki of 29 nM [55]. The aforementioned crystal structure of PDB: 5YJB with its specified reference was carefully chosen for our docking study due to the similarity of its 3D ligand core structure to our compounds specifically meaning the three rings around a central heterocyclic ring core. This reference ligand showed a tight binding via the interaction with a network of hydrogen binding with Lys661 and Asp 555. It was reported that it was located above the plane of the isoalloxazine ring of FAD [56] (Figure 6a).

The isoalloxazine ring of FAD was fitted into the large hydrophobic region, forming multiple H-bond interactions with Val 881, Val 333, and Met 332 and a p-H interaction with Trp751 (Figure 6b) [56,57].

#### 2.2.8. Molecular Docking Study

The investigated triazoles **22** and **23** (LSD_1_ inhibition IC_50_ = 46.35 nM, 65.11 nM) and the inactive triazole **18** IC_50_ = 1491 nM were docked into the active site of LSD_1_ and the results were tabulated on Table 6.

Compound **22** is well positioned to interact with one of its sulfur atoms with the most critically important amino acid, Lys661, for LSD1 inhibition [58], in addition to Asp 660 amino acid residue with hydrogen bonding interaction. It has also very similar binding patterns with surrounding residues, such as FAD interactions, which is illustrated as follows: one of the triazole rings and aromatic rings *via* arene–arene interaction with Try751 amino acid, in addition to arene–cationic interactions with Gly330 and Ala331, retains good stacking against the shelf-like hydrophobic surface. Moreover, hydrogen bonding interactions was observed in Arg316 via a second sulfur atom. The binding energy of the compound **22** was estimated by −17.15 Kcal/mol (Figure 7a).

Compound **23** is well positioned to interact with an additional critical amino acid residue for the LSD_1_ inhibition of Asp 555 via hydrogen bonding interaction with an *N*-triazole atom, with a binding energy of −15.94 Kcal/mol in a similar manner to a cocrystallized ligand (Table 6). Additional hydrogen bonding interaction was observed by a sulfur atom to Asp553 amino acid (Figure 7b). On the contrary, compound **18** lacks any binding with any key amino acids important for LSD1 inhibition, which may interpret its failure to have any LSD1 inhibition activity (Figure 7c). Moreover, the synthesized 1,3,4-thiadiazole **27** LSD1 inhibition with IC_50_ = 46.35 nM has shown effective binding to key amino acid Asp 555 essential for inhibition via hydrogen bonding interaction with a terminal amino group on a thiadiazole ring in addition to arene binding to a Try 761 residue indicating good fit, such as a FAD molecule (Figure 7d) (Table 6).

#### 2.2.9. Surface Mapping

The physicochemical features of the substituent group had an effect on the activity of the compounds. Hydrophobicity, in particular, was discovered to be linked to anticancer action. A surface map of the active region inside the LSD1 receptor was downloaded from the Protein Data Bank website, illustrating the active site’s hydrophobicity. Further investigations were conducted, and compound **22** was discovered in a hydrophobic surface mapping investigation (pink) occupying almost the same region with a cocrystallized ligand (cyan) (Figure 8). 

In addition, compound **22** displayed stronger lipophilic characteristics (greener patches) and was therefore able to make more contact with the enzyme’s lipophobic pocket (Figure 9a). Moreover, compound **22**,such as the reference compound GSK (Figure 9d), has a more flat shape, allowing the chemical to adopt a conformation that makes use of a more hydrophobic area inside the pocket. On the contrary, compound **18** surface mapping (Figure 9c) showed a distinct difference from compounds **22** and **27** (Figure 9b) and the reference compound in lacking sufficient hydrophobicity required for bonding and LSD1 inhibition. The results of the hydrophobic mapping and conformations show that hydrophobicity plays a specific function in the putative protein-binding location. 

#### 2.2.10. Computational Pharmacokinetic Analysis

The success of a drug candidate is determined not only by its good potential but also by a satisfactory ADME profile. Because a wide variety of experimental media and high-throughput in vitro ADME screens are available, it has the capacity to predict some important properties in silico and is valuable for the analysis of the good qualities of the molecules. Nowadays, it is recommended that the employment of computational ADME should be done as early as possible in the drug discovery process so as to reduce the number of safety issues [59]. Lipinski’s rule of five is helpful in describing molecular properties of a drug candidate, which is necessary to evaluate the important pharmacokinetic parameters, such as ADME. The rule is helpful in the drug design and development of a potential drug molecule [59,60]. The search engine further gave a compiled result on the lipophilicity and hydrophilicity of these molecules. Log P, a measure of the lipophilicity of a molecule, is the logarithm of the ratio of the concentration of a drug substance in two solvents in a unionized form. The aqueous solubility of a compound significantly affects its absorption and distribution characteristics. On the other hand, low water solubility often leads to bad absorption, and therefore, the general aim is to avoid poorly soluble compounds. Although the compound **27** exhibited a good hydrophilic–lipophilic balance and the same predicted bioavailability (Table 7), the **27** derivative with high lipophilicity was expected to show decent GI absorption. In addition, we calculated the total polar surface area (TPSA) since it is another key property that is related to drug bioavailability. Thus, passively absorbed molecules with TPSA > 140 are thought to have low oral bioavailability, which is quite perfect for our compound **27**.

It is also clear that our compounds **27** and **18** cannot be transferred into a CNS system compared with a reference drug that can permeate through the blood–brain barrier (BBB) and be affected by P-glycoprotein. In the present study, the synthesized ligand **27** was found to be in good agreement with the given criteria and can be said to possess good oral bioavailability.

## 3. Methods and Materials

### 3.1. General Chemistry

All reagents and solvents used were of the highest quality of analytical reagent grade and were used without further purification. Fine chemicals and solvents were purchased from BDH Chemicals Ltd. and Sigma-Aldrich. Melting points were measured on a Stuart Scientific SMP1 and are uncorrected. The progress of the reactions and purity of the synthesized compounds were determined by thin layer chromatography (TLC) using UV fluorescent silica gel Merck 60 F254 plates, and the spots were visualized using a UV lamp (254 nm). A PerkinElmer 1430 series FTIR spectrometer was used for the identification of functional groups in the range of 400–4000 cm^−1^. The NMR spectra were run with a Bruker spectrometer (400 MHz) with TMS as an internal reference. Elemental analyses were performed using a GmbH-Vario EL III Elementar Analyzer.


**Synthesis and characterization of benzene-1,3,5-tricarbohydrazide (11)**


A solution of triethyl benzene-1,3,5-tricarboxylate (**10**) (10 mmol) and hydrazine hydrate (45 mmol) in ethanol (50 mL) was refluxed for 4 h until the consumption of the starting material (as monitored using TLC). After cooling to room temperature, ethanol was removed under reduced pressure, and the resulting solid was recrystallized from ethanol to afford white powder. Yield: 94%; mp: 301–302 °C (lit. mp: 300 °C [61]); Rf = 0.42 (chloroform: methanol, 5:2). IR (KBr, υ): 3250–3390 (NH, NH_2_), 3080 (C-H_ar_), 1690 cm^−1^ (C=O). ^1^H NMR (400 MHz, DMSO-*d_6_*): δ_H_ 4.59 (6H, s, 3 × NH_2_), 8.33 (3H, s, 3 × H_arom_), 9.86 (3H, s, 3 × NH). ^13^C NMR (100 MHz, DMSO-*d_6_*): δ_C_ 128.51, 134.31 (C_arom_); 165.47 (3 × C=O). Anal. Calcd. for C_9_H_12_N_6_O_3_: C, 42.86; H, 4.80; N, 33.32. Found: C, 42.72; H, 4.74; N, 33.40.


**General procedure for the synthesis of acid thiosemicarbazides 12–17**


A mixture of benzene-1,3,5-tricarbohydrazide (**11**) (10 mmol) and the appropriate isothiocyanate derivatives (30 mmol) in ethanol (50 mL) was heated under reflux for 6 h until the consumption of the starting material as monitored by TLC. The reaction mixture was cooled; the obtained precipitate was filtered and recrystallized from ethanol to afford the targeted acid thiosemicarbazides **12**–**17**.


***Characterization of 2,2′,2″-(benzene-1,3,5-tricarbonyl)tris(N-methylhydrazine-1-carbothioamide)* (12)**


White powder, yield 90%; mp 202–203 °C; Rf = 0.28 (chloroform: methanol, 5:2). IR (KBr, υ): 3270–3380 (NH), 3080 (C-H_ar_), 2930 (C-H_al_), 1680 (C=O), 1280 cm^−1^ (C=S). ^1^H NMR (400 MHz, DMSO-*d_6_*): δ_H_ 3.39 (9H, s, 3 × CH_3_), 8.15 (3H, s, 3 × H_arom_), 8.60 (3H, s, 3 × NH), 9.46 (3H, s, 3 × NH), 10.54 (3H, s, 3 × NH). ^13^C NMR (100 MHz, DMSO-*d_6_*): δ_C_ 31.44 (3 × CH_3_); 130.84, 133.34 (C_arom_); 165.59 (3 × C=O); 182.70 (3 × C=S). Anal. Calcd. For C_15_H_21_N_9_O_3_S_3_: C, 38.21; H, 4.49; N, 26.73. Found: C, 38.33; H, 4.41; N, 26.77.


***Characterization of**2,2′,2″-(benzene-1,3,5-tricarbonyl)tris(N-ethylhydrazine-1-carbothioamide)* (13)**


White powder, yield 89%; mp 209–210 °C; Rf = 0.27 (chloroform: methanol, 5:2). IR (KBr, υ): 3260–3370 (NH), 3070 (C-H_ar_), 2960 (C-H_al_), 1690 (C=O), 1290 cm^−1^ (C=S). ^1^H NMR (400 MHz, DMSO-*d_6_*): δ_H_ 1.07–1.10 (9H, m, 3 × CH_3_), 3.45–3.52 (6H, m, 3 × CH_2_), 8.19 (2.60H, s, H_arom_), 8.32 (0.40H, s, H_arom_), 8.61 (3H, s, 3 × NH), 9.39 (2.80H, s, NH), 9.86 (0.45H, s, NH), 10.51 (2.75H, s, NH). ^13^C NMR (100 MHz, DMSO-*d_6_*): δ_C_ 14.98 (3 × CH_3_); 39.00 (3 × CH_2_); 128.52, 130.81, 133.37, 134.29 (C_arom_); 165.48 (3 × C=O); 181.72 (3 × C=S). Anal. Calcd. for C_18_H_27_N_9_O_3_S_3_: C, 42.09; H, 5.30; N, 24.54. Found: C, 42.35; H, 5.23; N, 24.42.


***Characterization of**2,2′,2″-benzenetricarbonyltris(N-cyclohexylhydrazine carbothioamide)* (14)**


White powder, yield 87%; mp 190–191 °C; Rf = 0.25 (chloroform: methanol, 5:2). IR (KBr, υ): 3240–3365 (NH), 3080 (C-H_ar_), 2920 (C-H_al_), 1695 (C=O), 1295 cm^−1^ (C=S). ^1^H NMR (400 MHz, DMSO-*d_6_*): δ_H_ 1.05–1.08 (6H, m, 3 × CH_2_), 1.27–1.33 (12H, m, 6 × CH_2_), 1.57–1.83 (12H, m, 6 × CH_2_), 4.35–4.39 (3H, m, 3 × CH), 8.33–8.68 (6H, m, 3 × H_arom_, 3 × NH), 9.28 (3H, bs, 3 × NH), 9.84 (1H, s, NH), 10.03 (2H, s, 2 × NH). ^13^C NMR (100 MHz, DMSO-*d_6_*): δ_C_ 25.35, 25.63, 32.44 (15 × CH_2_); 56.50 (3 × CH); 128.51, 130.36, 130.57, 130.68, 134.31, 134.67 (C_arom_); 164.87, 165.47, 165.81 (3 × C=O, 3 × C=S). Anal. Calcd. for C_30_H_45_N_9_O_3_S_3_: C, 53.31; H, 6.71; N, 18.65. Found: C, 53.44; H, 6.75; N, 18.58.


***Characterization of 2,2′,2″-(benzene-1,3,5-tricarbonyl)tris(N-allylhydrazine-1-carbothioamide)* (15)**


White powder, yield 88%; mp 215–216 °C; Rf = 0.24 (chloroform: methanol, 5:2). IR (KBr, υ): 3250–3390 (NH), 3090 (C-H_ar_), 2900 (C-H_al_), 1680 (C=O), 1280 cm^−1^ (C=S). ^1^H NMR (400 MHz, DMSO-*d_6_*): δ_H_ 4.13 (6H, s, 3 × NHCH_2_), 5.07 (3H, d, *J* = 8 Hz, 3 × =CH), 5.17 (3H, d, *J* = 16 Hz, 3 × =CH), 5.81–5.87 (3H, m, 3 × CH_2_CH=), 8.37 (3H, bs, 3 × H_arom_), 8.61 (3H, s, 3 × NH), 9.50 (3H, s, 3 × NH), 10.57 (3H, s, 3 × NH). ^13^C NMR (100 MHz, DMSO-*d_6_*): δ_C_ 46.35 (3 × NHCH_2_); 115.89, 130.86, 133.35, 135.42 (C=C, C_arom_); 165.53 (3 × C=O); 190.14 (3 × C=S). Anal. Calcd. for C_21_H_27_N_9_O_3_S_3_: C, 45.89; H, 4.95; N, 22.93. Found: C, 45.80; H, 4.88; N, 22.86.


***Characterization of**2,2′,2″-(benzene-1,3,5-tricarbonyl)tris(N-benzylhydrazine-1-carbothioamide)* (16)**


White powder, yield 87%; mp 176–177 °C; Rf = 0.22 (chloroform: methanol, 5:2). IR (KBr, υ): 3230–3370 (NH), 3070 (C-H_ar_), 2930 (C-H_al_), 1690 (C=O), 1300 cm^−1^ (C=S). ^1^H NMR (400 MHz, DMSO-*d_6_*): δ_H_ 4.67 and 4.68 (2H, 2s, CH_2_), 4.81 (2H, s, CH_2_), 4.94 (2H, s, CH_2_), 7.24–7.46 (18H, m, 3 × C_6_H_5_ and 3 × NH), 8.68 (1.5H, s, H_arom_), 8.80 (1.5H, bs, H_arom_), 9.63 (3H, s, 3 × NH), 10.68 (3H, s, 3 × NH). ^13^C NMR (100 MHz, DMSO-*d_6_*): δ_C_ 65.96, 66.70 (3 × CH_2_); 127.11, 127.46, 127.49, 127.54, 127.64, 127.82, 127.87, 128.54, 128.69, 128.75, 128.84, 129.35, 130.98, 133.37, 135.19, 138.37, 138.74, 139.79 (C_arom_); 165.68 (3 × C=O); 188.61, 190.76 (3 × C=S). Anal. Calcd. for C_33_H_33_N_9_O_3_S_3_: C, 56.63; H, 4.75; N, 18.01. Found: C, 56.43; H, 4.70; N, 18.11.


***Characterization of 2,2′,2″-(benzene-1,3,5-tricarbonyl)tris(N-phenylhydrazine-1-carbothioamide)* (17)**


White powder, yield 90%; mp 163–165 °C; Rf = 0.20 (chloroform: methanol, 5:2). IR (KBr, υ): 3250–3390 (NH), 3080 (C-H_ar_), 1680 (C=O), 1285 cm^−1^ (C=S). ^1^H NMR (400 MHz, DMSO-*d_6_*): δ_H_ 7.16–7.43 (15H, m, 3 × C_6_H_5_), 8.71–8.74 (3H, m, 3 × H_arom_), 9.93 (6H, bs, 6 × NH), 10.77–10.95 (3H, 3s, 3 × NH). ^13^C NMR (100 MHz, DMSO-*d_6_*): δ_C_ 125.72, 126.74, 128.55, 133.40, 133.50, 133.95, 133.98, 139.64 (C_arom_); 165.37, 165.53, 165.79 (3 × C=O); 181.41, 181.52 (3 × C=S). Anal. Calcd. for C_30_H_27_N_9_O_3_S_3_: C, 54.78; H, 4.14; N, 19.16. Found: C, 54.86; H, 4.18; N, 19.10.


**General procedure for the synthesis of 1,2,4-triazole-3-thiones 18–23**


The appropriate acid thiosemicarbazide **12**–**17** (1 mmol) was dissolved in 10% aqueous solution of sodium hydroxide (30 mL). The mixture was allowed to react at reflux for 6 h, cooled, filtered, and acidified with hydrochloric acid. The crude product was collected by filtration and recrystallized from ethanol to afford the targeted 1,2,4-triazoles **18**–**23**.


***Characterization of 3,3′,3″-(benzene-1,3,5-triyl)tris(4-methyl-1H-1,2,4-triazole-5(4H)-thione)* (18)**


White powder, yield 88%; mp 279–281 °C; Rf = 0.69 (chloroform: methanol, 5:2). IR (KBr, υ): 3320 (NH), 3060 (C-H_ar_), 2970 (C-H_al_), 1620 (C=N), 1300 cm^−1^ (C=S). ^1^H NMR (400 MHz, DMSO-*d_6_*): δ_H_ 3.59 and 3.61 (9H, 2s, 3 × CH_3_), 8.25 (1.25H, s, H_arom_), 8.33 (0.50H, s, H_arom_), 8.40 (1.25H, s, H_arom_), 14.07, and 14.09 (3H, 2s, 3 × NH). ^13^C NMR (100 MHz, DMSO-*d_6_*): δ_C_ 32.11, 32.13 (3 × CH_3_); 127.92, 128.20, 130.85, 131.55, 132.49, 133.00 (C_arom_); 150.49, 150.57 (C=N); 166.26, 168.32, 168.35 (C=S). Anal. Calcd. for C_15_H_15_N_9_S_3_: C, 43.15; H, 3.62; N, 30.19. Found: C, 43.30; H, 3.69; N, 30.32.


***Characterization of 3,3′,3″-(benzene-1,3,5-triyl)tris(4-ethyl-1H-1,2,4-triazole-5(4H)-thione)* (19)**


White powder, yield 87%; mp 266–268 °C; Rf = 0.68 (chloroform: methanol, 5:2). IR (KBr, υ): 3340 (NH), 3080 (C-H_ar_), 2930 (C-H_al_), 1630 (C=N), 1290 cm^−1^ (C=S). ^1^H NMR (400 MHz, DMSO-*d_6_*): δ_H_ 1.17–1.22 (9H, m, 3 × CH_3_), 4.06–4.16 (6H, m, 3 × CH_2_), 8.23 (0.75H, s, H_arom_), 8.27 (0.75H, s, H_arom_), 8.38 (1.50H, s, H_arom_), 14.06, and 14.09 (3H, 2s, 3 × NH). ^13^C NMR (100 MHz, DMSO-*d_6_*): δ_C_ 13.83, 13.85 (3 × CH_3_); 39.75, 39.81 (3 × CH_2_); 128.05, 128.46, 131.12, 131.65, 132.61, 133.17 (C_arom_); 149.94, 150.08 (C=N); 166.16, 167.68 (C=S). Anal. Calcd. for C_18_H_21_N_9_S_3_: C, 47.04; H, 4.61; N, 27.43. Found: C, 47.18; H, 4.68; N, 27.55.


***Characterization of 3,3′,3″-(benzene-1,3,5-triyl)tris(4-cyclohexyl-1H-1,2,4-triazole-5(4H)-thione)* (20)**


White powder, yield 85%; mp 251–252 °C; Rf = 0.65 (chloroform: methanol, 5:2). IR (KBr, υ): 3310 (NH), 3050 (C-H_ar_), 2910 (C-H_al_), 1615 (C=N), 1300 cm^−1^ (C=S). ^1^H NMR (400 MHz, DMSO-*d_6_*): δ_H_ 0.88–0.94 (4H, m, 2 × CH_2_), 1.15–1.21 (6H, m, 3 × CH_2_), 1.54–1.99 (20H, m, 10 × CH_2_), 4.32–4.35 (3H, m, 3 × CH), 8.06 (0.50H, s, H_arom_), 8.09 (1H, s, H_arom_), 8.28 (1H, s, H_arom_), 8.65 (0.50H, s, H_arom_), 14.00, and 14.03 (3H, 2s, 3 × NH). ^13^C NMR (100 MHz, DMSO-*d_6_*): δ_C_ 25.14, 25.16, 25.92, 26.02, 30.36, 30.72 (15 × CH_2_); 57.66 (3 × CH); 128.00, 128.82, 132.36, 132.41, 132.46, 133.01, 134.05, 134.36, 134.54, 135.54 (C_arom_); 149.88, 150.23 (C=N); 166.12, 166.22, 166.33, 167.03 (C=S). Anal. Calcd. for C_30_H_39_N_9_S_3_: C, 57.94; H, 6.32; N, 20.27. Found: C, 57.80; H, 6.27; N, 20.38.


***Characterization of 3,3′,3″-(benzene-1,3,5-triyl)tris(4-allyl-1H-1,2,4-triazole-5(4H)-thione)* (21)**


White powder, yield 87%; mp 287–288 °C; Rf = 0.64 (chloroform: methanol, 5:2). IR (KBr, υ): 3280 (NH), 3070 (C-H_ar_), 2930 (C-H_al_), 1620 (C=N), 1290 cm^−1^ (C=S). ^1^H NMR (400 MHz, DMSO-*d_6_*): δ_H_ 4.76 (6H, bs, 3 × NHCH_2_), 4.85 (3H, d, *J* = 16 Hz, 3 × =CH), 5.11 (3H, d, *J* = 8 Hz, 3 × =CH), 5.76–5.85 (3H, m, 3 × CH_2_CH=), 8.16 (3H, s, 3 × H_arom_), 14.20 (3H, s, 3 × NH). ^13^C NMR (100 MHz, DMSO-*d_6_*): δ_C_ 46.35 (3 × NHCH_2_); 117.87, 128.13, 130.63, 131.98 (C=C, C_arom_); 150.15 (C=N); 168.25 (C=S). Anal. Calcd. for C_21_H_21_N_9_S_3_: C, 50.89; H, 4.27; N, 25.43. Found: C, 50.77; H, 4.31; N, 25.35.


***Characterization of 3,3′,3″-(benzene-1,3,5-triyl)tris(4-benzyl-1H-1,2,4-triazole-5(4H)-thione)* (22)**


White powder, yield 85%; mp 240–241 °C; Rf = 0.62 (chloroform: methanol, 5:2). IR (KBr, υ): 3315 (NH), 3040 (C-H_ar_), 2945 (C-H_al_), 1615 (C=N), 1310 cm^−1^ (C=S). ^1^H NMR (400 MHz, DMSO-*d_6_*): δ_H_ 6.71 and 6.74 (6H, 2s, 3 × CH_2_), 8.28–8.87 (18H, m, H_arom_), 12.90, 13.13 and 13.29 (3H, 3s, 3 × NH). ^13^C NMR (100 MHz, DMSO-*d_6_*): δ_C_ 67.22, 67.28 (3 × CH_2_); 124.70,126.68, 126.82, 127.70,128.60, 130.27, 130.30, 134.25, 138.66, 138.75, 144.61, 145.16, 147.18, 147.34, 147.77, 147.85, 148.11 (C_arom_); 159.70, 160.05 (C=N); 190.30, 190.51 (C=S). Anal. Calcd. for C_33_H_27_N_9_S_3_: C, 61.37; H, 4.21; N, 19.52. Found: C, 61.56; H, 4.29; N, 19.41.


***Characterization of 3,3′,3″-(benzene-1,3,5-triyl)tris(4-phenyl-1H-1,2,4-triazole-5(4H)-thione)* (23)**


White powder, yield 87%; mp 229–230 °C; Rf = 0.60 (chloroform: methanol, 5:2). IR (KBr, υ): 3300 (NH), 3070 (C-H_ar_), 1620 (C=N), 1290 cm^−1^ (C=S). ^1^H NMR (400 MHz, DMSO-*d_6_*): δ_H_ 7.10–7.30 (5H, m, H_arom_), 7.42–7.58 (10H, m, H_arom_), 7.80 (2H, s, H_arom_), 8.07 (0.70H, s, H_arom_), 8.44 (0.30H, s, H_arom_), 14.16, 14.21, and 14.25 (3H, 3s, 3 × NH). ^13^C NMR (100 MHz, DMSO-*d_6_*): δ_C_ 127.06, 127.15, 127.28, 128.86, 129.02, 129.13, 129.95, 130.08, 130.75, 131.60, 131.98, 132.09, 132.20, 133.03, 134.12, 134.48, 134.72 (C_arom_); 149.03, 149.32, 149.55 (C=N); 165.68, 166.05, 169.20, 169.30, 169.39 (C=S). Anal. Calcd. for C_30_H_21_N_9_S_3_: C, 59.68; H, 3.51; N, 20.88. Found: C, 59.80; H, 3.45; N, 20.80.


**General procedure for the synthesis of 1,3,4-thiadiazole-3-thiones 24–28**


A solution of acid thiosemicarbazide **12**–**15** and **17** (1 mmol) in cold concentrated sulfuric acid (30 mL) was stirred for 30 min at 0 °C. Then, the mixture was allowed to reach room temperature. After stirring for an additional 16 h, the resulting solution was poured into ice-cold water and treated with ammonium hydroxide solution until pH was adjusted to 8. The resulting product was filtered, washed with water, dried, and recrystallized from ethanol, yielding the targeted thiadiazoles **24**–**28**.


***Characterization of 5,5′,5″-(benzene-1,3,5-triyl)tris(N-methyl-1,3,4-thiadiazol-2-amine)* (24)**


Off-white powder, yield 85%; mp 247–248 °C; Rf = 0.59 (chloroform: methanol, 5:2). IR (KBr, υ): 3280 (NH), 3050 (C-H_ar_), 2900 (C-H_al_), 1610 cm^−1^ (C=N). ^1^H NMR (400 MHz, DMSO-*d_6_*): δ_H_ 2.97 (7H, s, CH_3_), 3.93 (2H, s, CH_3_), 7.12 (3H, bs, 3 × NH), 8.07 (0.5H, s, H_arom_), 8.09 (1H, bs, H_arom_), 8.23 (1H, s, H_arom_), 8.24 (0.5H, bs, H_arom_). ^13^C NMR (100 MHz, DMSO-*d_6_*): δ_C_ 31.89 (CH_3_); 53.17 (CH_3_); 124.31, 127.29, 127.51, 131.89, 132.76, 132.99 (C_arom_); 154.33, 154.60, 165.50, 170.35, 170.39 (C=N). Anal. Calcd. for C_15_H_15_N_9_S_3_: C, 43.15; H, 3.62; N, 30.19. Found: C, 43.25; H, 3.69; N, 30.28. 


***Characterization of 5,5′,5″-(benzene-1,3,5-triyl)tris(N-ethyl-1,3,4-thiadiazol-2-amine)* (25)**


Off-white powder, yield 84%; mp 233–234 °C; Rf = 0.57 (chloroform: methanol, 5:2). IR (KBr, υ): 3310 (NH), 3080 (C-H_ar_), 2930 (C-H_al_), 1600 cm^−1^ (C=N). ^1^H NMR (400 MHz, DMSO-*d_6_*): δ_H_ 1.23 (9H, t, *J* = 4 Hz, 3 × CH_3_), 3.93–3.98 (6H, m, 3 × CH_2_), 6.99 (1H, s, NH), 7.11 (1H, s, NH), 7.24 (1H, s, NH), 8.05 (0.5H, s, H_arom_), 8.22 (2H, s, H_arom_), 8.25 (0.5H, bs, H_arom_). ^13^C NMR (100 MHz, DMSO-*d_6_*): δ_C_ 14.64 (3 × CH_3_); 53.18 (3 × CH_2_); 124.41, 127.35, 127.54, 131.90, 132.69, 132.91 (C_arom_); 154.18, 154.43, 165.49, 169.37, 169.42 (C=N). Anal. Calcd. for C_18_H_21_N_9_S_3_: C, 47.04; H, 4.61; N, 27.43. Found: C, 47.18; H, 4.68; N, 27.33.


***Characterization of 5,5′,5″-(benzene-1,3,5-triyl)tris(N-cyclohexyl-1,3,4-thiadiazol-2-amine)* (26)**


Off-white powder, yield 83%; mp 218–219 °C; Rf = 0.56 (chloroform: methanol, 5:2). IR (KBr, υ): 3290 (NH), 3020 (C-H_ar_), 2960 (C-H_al_), 1600 cm^−1^ (C=N). ^1^H NMR (400 MHz, DMSO-*d_6_*): δ_H_ 0.83–0.94 (4H, m, 2 × CH_2_), 1.19–1.39 (12H, m, 6 × CH_2_), 1.57–2.03 (14H, m, 7 × CH_2_), 3.52–3.58 (3H, m, 3 × CH), 7.59–7.63 (3H, m, 3 × NH), 8.04–8.26 (3H, m, H_arom_). ^13^C NMR (100 MHz, DMSO-*d_6_*): δ_C_ 24.75, 25.14, 25.76, 30.76, 31.78, 32.49 (15 × CH_2_); 54.42 (3 × CH); 124.31, 132.98 (C_arom_); 154.25, 155.18, 168.58, 169.63, (C=N). Anal. Calcd. for C_30_H_39_N_9_S_3_: C, 57.94; H, 6.32; N, 20.27. Found: 57.78; H, 6.25; N, 20.17.


***Characterization of 5,5′,5″-(benzene-1,3,5-triyl)tris(N-allyl-1,3,4-thiadiazol-2-amine)* (27)**


Off-white powder, yield 84%; mp 259–260 °C; Rf = 0.54 (chloroform: methanol, 5:2). IR (KBr, υ): 3320 (NH), 3030 (C-H_ar_), 2940 (C-H_al_), 1620 cm^−1^ (C=N). ^1^H NMR (400 MHz, DMSO-*d_6_*): δ_H_ 4.70–4.77 (6H, m, 3 × NHCH_2_), 4.85–4.91 (3H, m, 3 × =CH), 5.19 (3H, d, *J* = 8 Hz, 3 × =CH), 5.84–5.91 (3H, m, 3 × CH_2_CH=), 7.25 (3H, bs, 3 × NH), 8.09 (1.5H, bs, H_arom_), 8.21 (1H, s, H_arom_), 8.26 (0.5H, bs, H_arom_). ^13^C NMR (100 MHz, DMSO-*d_6_*): δ_C_ 50.48, 52.90 (3 × NHCH_2_); 124.21, 127.21, 127.65, 131.56, 132.34, 132.63, 132.94 (C=C, C_arom_); 154.14, 154.36, 154.40, 154.61, 165.26, 166.40, 169.63, 169.67 (C=N). Anal. Calcd. for C_21_H_21_N_9_S_3_: C, 50.89; H, 4.27; N, 25.43. Found: C, 50.70; H, 4.33; N, 25.32.


***Characterization of 5,5′,5″-(benzene-1,3,5-triyl)tris(N-phenyl-1,3,4-thiadiazol-2-amine)* (28)**


Off-white powder, yield 85%; mp 204–205 °C; Rf = 0.51 (chloroform: methanol, 5:2). IR (KBr, υ): 3280 (NH), 3050 (C-H_ar_), 1610 cm^−1^ (C=N). ^1^H NMR (400 MHz, DMSO-*d_6_*): 7.12–7.33 (8H, m, 3 × NH, H_arom_), 7.62–7.67 (10H, m, H_arom_). ^13^C NMR (100 MHz, DMSO-*d_6_*): δ_C_ 127.23, 127.31, 128.89, 128.95, 129.34, 129.89, 130.11, 130.89, 131.78, 132.58, 132.83, 133.43, 134.52, 134.86, 135.09, 149.67, 149.91 (C_arom_); 154.89, 155.67, 166.87, 169.78, 167.18 (C=N). Anal. Calcd. for C_30_H_21_N_9_S_3_: C, 59.68; H, 3.51; N, 20.88. Found: C, 59.51; H, 3.57; N, 20.76.

### 3.2. Cytotoxicity Assay

The in vitro cytotoxic testing of novel derivatives was performed against different three cancer cell lines, HepG2, MCF-7, and HCT116, using standard MTT assay as previously reported [22,62,63,64].

### 3.3. LSD_1_ Enzymatic Inhibitory Activity Assay 

The inhibition of LSD_1_ activity by our tested compounds was evaluated according to a previously reported procedure [65,66,67,68] and as mentioned in the assay kit in Appendix A. 

### 3.4. Cell Cycle Analysis and Apoptosis Detection 

Using the annexin V-FITC and PI apoptosis kit (eBioscience^TM^, San Diego, CA, USA), the apoptosis detection was performed. MCF cells were plated onto a six-well plate at a 600,000 cells/mL density and then treated with compound **27** at 10 Mg/mL after 24 h of incubation, and the assay was completed as previously described [38,69].

### 3.5. Apoptosis Assay

The MCF-7 cells were treated with tested compounds **27** for 24 h and stained with annexin V-FITC/PI to detect externalization of PS from cell membrane and analyze apoptotic and necrotic cell population as previously described [70,71]. 

### 3.6. DNA Fragmentation

DNA fragmentation was quantitatively determined using a diphenylamine (DPA) reagent according to the method of Boraschi and Maurizi [72,73,74].

### 3.7. Cell Cycle Analysis

Cell cycle arrest and distribution was assessed, using the Propidium Iodide Flow Cytometry Kit (ab139418, Abcam, USA, CA, USA), followed by flow cytometry analysis according to the previous methodology [75].

### 3.8. MAO Activity Screening

The coupled assay approach was used to evaluate MAO activity in which the product, H_2_O_2_, acts with horseradish peroxidase to convert 10-acetyl-3,7-dihydroxyphenoxazine to the fluorescent resorufin. The percent inhibition was computed from the top and bottom of the resultant curve using the equation for a three-parameter curve in GraphPad Prism v. 4, (Graphstats, New York, NY, USA) [52,76,77,78].

### 3.9. Statistical Analysis 

All experiments were carried out in triplicate, and each data point represented the overall mean of at least two independent experiments. The findings were expressed as mean ± standard deviation (SD). The results were analyzed using GraphPad Prism (version 4, Graphstats, New York, NY, USA) and Microsoft Office Excel software to carry out the statistical analysis. One-way analysis of variance technique, followed by Tukey’s test, was applied to observe the significance between the groups. The entire statistical analysis was carried out at *p* ≤ 0.05.

### 3.10. Molecular Docking

MOE (Chemical Computing Group Inc. software 2014, Montreal, QC, Canada, accessed on 20 January 2014) was used to build the 3D structures of some selected substituted compounds in their neutral forms, which represented the best and worst LSD1 inhibitors. The lowest energy conformers of novel analogues ‘global-minima’ were docked into the binding pocket of PDB: 5YJB without the original ligand (8 WC) from Brookhaven National Laboratory’s Protein Data Bank. After adding the hydrogens, the enzyme structure was refined using a technique that gradually reduced and decreased the restrictions on the enzyme until the RMSD gradient was 0.01 kcal/mol A. The molecular mechanics force field ‘AMBER’ was used to perform energy minimization. Energy minimizations (EM) were carried out for each selected candidate using 1000 steps of steepest descent, followed by conjugate gradient minimization to an RMSD energy gradient of 0.01 Kcal/mol. The difference between the energy of the complex and the individual energies of the enzyme and ligand was used to compute the binding energy [59,79]. Molecular Operating Environment software (Montreal, QC, Canada, accessed on 20 January 2014) was used to perform a flexible alignment experiment on the chemicals under investigation (MOE of Chemical Computing Group Inc. on a Core i7 2.3 GHz workstation). A radius of 10.0 Å was used to detect the enzyme’s active site. The partial atomic charges for each analogue were assigned using the program’s semiempirical mechanical computation “AM1” method [80]. The conformational search was carried out. All of the conformers were reduced until the RMSD deviation was less than 0.01 Kcal/mol, and then surface mapping was performed, using the following color codes: pink, hydrogen bond; blue, mild polar.

## 4. Conclusions

LSD1 (KDM1A) is one of the most well-studied KDMs, with higher levels seen in a variety of malignancies [81]. As a result, LSD1 is a promising target for the development of cancer therapies. In this study, a series of aryl-centered trisubstituted systems enriched with a 1,2,4 triazole and 1,3,4-thiadiazole framework were designed, synthesized, and evaluated for cytotoxic activity using MTT assay against different three cancer cell lines. In addition, the design succeeded in introducing a novel scaffold, which was evaluated for LSD_1_ inhibitory activity. Most of the synthesized derivatives had potency with IC_50_ at a submicromolar value and selectivity performance comparable to reference inhibitors. Compound **27** with 1,3,4-thiadiazole core was the most promising potent LSD1 inhibitor with a IC_50_ value of 0.046 µM. Moreover, a detailed mechanistic analysis on the cancer cell lines revealed that when compared with the positive control medication, the majority of the compounds had stronger antiproliferative activity. In MCF-7 and HepG2 cancer cells, compound **27** also enhanced apoptosis, produced cell cycle arrest at the G2/M phase, and caused DNA fragmentation. Molecule 27 appears to offer a lot of potential as a novel multi-triazole-based lead compound for the identification of new anticancer medicines that target the LSD1 enzyme, based on our findings. We intend to keep optimizing this molecule in order to create chemical entities with great anticancer activity.

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
