# Peer review of "Design and Synthesis of Benzene Homologues Tethered with 1,2,4-Triazole and 1,3,4-Thiadiazole Motifs Revealing Dual MCF-7/HepG2 Cytotoxic Activity with Prominent Selectivity via Histone Demethylase LSD1 Inhibitory Effect"

_ijms, 2022, doi:10.3390/ijms23158796_

Round 1

Reviewer 1 Report

The article entitled "Design and Synthesis of Planar System Homologues Tethered with 1,2,4-Triazole and 1,3,4-Thiadiazole Motifs Revealing Potent MCF-7/HepG2 Cytotoxic activity with Prominent Selectivity via Histone Demethylase LSD1 Inhibitory effect" has been reviewed. 

The present work is appreciable and may be of great interest to readers. Each experiment and the related experimental method have been adequately and clearly described. In addition, the pharmacological results obtained are promising and in accordance with the authors' intended goals.

However, before the manuscript is suitable for publication, the authors should consider the following observations:

- The abstract should be shortened; it exceeds the 200-word guideline in the journal guidelines 

- The authors should include the statistical analysis in Table 3, Figure 2, and Table 4. Also, from Figure 5, the SD is not evident.  It would be important to include a subsection on statistical analysis in the Materials and Methods section, specifying the type of analysis performed to calculate significance. 

Author Response

Comments of Reviewer # 1:

The article entitled "Design and Synthesis of Planar System Homologues Tethered with 1,2,4-Triazole and 1,3,4-Thiadiazole Motifs Revealing Potent MCF-7/HepG2 Cytotoxic activity with Prominent Selectivity via Histone Demethylase LSD1 Inhibitory effect" has been reviewed.

The present work is appreciable and may be of great interest to readers. Each experiment and the related experimental method have been adequately and clearly described. In addition, the pharmacological results obtained are promising and in accordance with the authors' intended goals.

However, before the manuscript is suitable for publication, the authors should consider the following observations:

  1. The abstract should be shortened; it exceeds the 200-word guideline in the journal guidelines

Response: The abstract was rewritten to meet the journal 200 words.

  1. The authors should include the statistical analysis in Table 3, Figure 2, and Table 4. Also, from Figure 5, the SD is not evident. It would be important to include a subsection on statistical analysis in the Materials and Methods section, specifying the type of analysis performed to calculate significance.

Response: SD values were added for the values of figure 5 within the text and a subsection of statistical analysis was added in methods section.

Reviewer 2 Report

Please, see attached file

Author Response

Comments of Reviewer # 2:

The present manuscript entitled “Design and Synthesis of Planar System Homologues Tethered with 1,2,4-Triazole and 1,3,4-Thiadiazole Motifs Revealing Potent MCF-7/HepG2 Cytotoxic activity with Prominent Selectivity via Histone Demethylase LSD1 Inhibitory effect” is a very interesting and complete research work, demonstrating the potentiality of some novel N-derived compounds as LSD1 inhibitors. In general, although the results are promising and relevant for the pharmaceutical field, the manuscript quality is poor. Below, some of the points to be significantly improved are listed:

  1. Abstract must be clearly improved.

Response: The abstract was improved and reduced as journal needs.

  1. In general, the English grammar must be revised.

Response: all English grammar was revised and corrected using Grammarly premium version.

  1. Line 85. Could please authors explain how the introduction of a N atom provide flexibility to a 5- or 6-members cycle?

Response: Thanks for you, the flexibility of 6- or 5- membered ring gained by insertion of heteroatom as N comes from the ability of the ring to form polar interaction as hydrogen-bonding type in the molecular target.

  1. In general, schemes and figures should appear just after the end of the paragraph where they are mentioned for the first time.

Response: they reorganized in the manuscript.

  1. The IR spectra should also be included in the supplementary.

Response: As suggested by the learned reviewer, we are unable to provide the requested IR data in the supplementary section because we are away from our university for the summer vacation and all IR data are saved on the Lab's computer.

Figures in supplementary are not indicated in the manuscript.

Response: They were added into the manuscript.

  1. The NMR figures in the supplementary present the chemical shifts of the signals very small and are difficult to see. The intensity of these signals is also very small compared to the size of the figure. I would suggest increasing the intensity of the signals in all NMRs. In addition, each NMR should include the structure of its compound and an indication of the atoms that each signal corresponds to.

Response: This comment has been addressed in the revised version of the supplementary, as requested by the reviewer. All protons and carbons have already been assigned in the experimental section.

  1. The description of commercial assays should be removed from the supplementary

Response: ok it was removed from supplementary part.

  1. The NMR of compound 10 was not included in the supplementary. What is the reason for not including it? If it is because it was already published, authors have not included the reference describing it. Indeed, authors have not mentioned any reference used for any synthesis.

Response: Thank you for your insightful comment; compound 10 is a well-known compound that was prepared in accordance with reference 40, which was inserted as suggested on page 4. (See manuscript).

https://www.tandfonline.com/doi/citedby/10.1080/17518250903567261?scroll=top&needAccess=true

  1. The use of subscript and superscripts are missing in all cases where required.

Response: all manuscript was revised for all cases.

  1. Figure 2. Both figures should include what axe y represents and the legend describing the different colors

Response: Both figures were modified and y axis was defined. Two different colors refer to the cell lines used.

  1. Figure 5. SD values should be included in the bars.

Response: The SD values were added to figure.

  1. The footnote of table 1 is misconfigured. The acronym ND has not been defined.

Response: The footnote of the Table 1 was modified to be clear.

  1. Footnote of figure 6 is misconfigured.

Response: The footnote legend of the figure was modified to be clear.

Author Response

Comments of Reviewer # 3:

In the manuscript titled:”Design and Synthesis of Planar System Homologues Tethered with 1,2,4-Triazole and 1,3,4-Thiadiazole Motifs Revealing PotentMCF-7/HepG2 Cytotoxic activity with Prominent Selectivity via Histone Demethylase LSD1 Inhibitory effect”, the authors report the synthesis, the characterization and the bioactivity on breast cancer (MCF-7), colon cancer (HTC116) and liver cancer (Hep-G2) cell lines of 6 derivatives of 2,4-dihydro-3H-1,2,4-triazole-3-thione and 5 derivatives of 2-amino-1,3,4-thiadiazole. In addition the authors have performed docking calculation on three compounds selected among the whole series.

Although the topic could be of some interest in the field of medicinal chemistry, it doesn’t fit exactly the purpose of the special issue, which regards malignant melanoma. In this study were used cell lines not belonging to melanoma and the authors have not discussed anywhere in the manuscript about the role of LSD1 in melanoma. In addition, the manuscript presents many weakness.

Response:

  1. The title of manuscript should be more concise. It is too long and misleading for two reason:
  2. a) because the cytotoxicity to be considerer potent needs to have sub-micromolar values
  3. b) Because the triazoltione or tiadiazole units are not coplanar with the benzene ring, but shows adihedral angle around 125-126°, making the whole system non-planar.

Response: The title was modified according to recommendations.

  1. The abstract is too long (337 word!) and needs to be rewritten. In guide to the authors is reported:“The abstract should be a total of about 200 words maximum”.

Response: The abstract was rewritten to meet the journal 200 words.

  1. At row 49 the numbers 11, 13 and 18 as well as at row 54 the numbers 18, 14 and 13 should put inbold.

Response: ok made bold.

  1. In the abstract, at row 84 and following, instead to use 1,2,4-triazole derivatives is better to use 1,2,4-triazole-3-thione

Response: ok it was replaced.

  1. Introduction needs to be improved in all its part. The reported state of art of lysine demethylases incancer is too synthetic and needs to be added of further very recent literature this topic. See:
  2. a) Punnia-Moorthy G, Hersey P, Emran AA and Tiffen J (2021) “Lysine Demethylases: Promising DrugTargets in Melanoma and Other Cancers”. Front. Genet. 12:680633. doi:10.3389/fgene.2021.680633).
  3. b) Junmin Lee, Thomas G. Molley, Christopher H. Seward , Amr A. Abdeen, Huimin Zhang,

Xiaochun Wang, Hetvi Gandhi, Jia-Lin Yang, Katharina Gaus, Kristopher A. Kilian, (2020) “Geometricregulation of histone state directs melanoma reprogramming” COMMUNICATIONS BIOLOGY 3:341,https://doi.org/10.1038/s42003-020-1067-1.

Response: the introduction part was modified with insertion of the two recent references and others for explaining the role of LSD1 in melanoma and cancer types.

  1. At row 106 the authors report:”…this work aims to the synthesis of a library of new aromatic…”, butcompounds 18, 19 (CAS Number: 929714-18-1) and 21 (CAS Number: 929714-19-2) are not newbecause already reported in the patent EP1764650

Response:

Herein, we use 1,3,5-benzenetricarboxylic acid as key precursor for obtaining these products instead of 1,3,5-pentanetricarboxylic acid which was usedwith refluxing concentrated sulfuric acid as mentioned in the patent. Thus, these derivatives 18, 19, 20 were assessed using new method. In addition, these derivatives were investigated in the field of photosensitive composition and lithographic printing plate precursor for image-recording method which are completely different from our rational in the medical field. Therefore, we consider them new from synthetic and medical use points of view. To be more accurate, we modified the sentence by adding the word ‘some’ before new compounds.

  1. All the reported compounds number should be in bold.

Response:The compound numbers were rechecked throughout the manuscript to be bold.

  1. At paragraph 2.1, the description is tedious, because all the reaction are simple functional grouptransformations, so the authors should improve this part.

Response:Paragraph 2.1 has been reconsidered and modified to be more concise.

  1. In the chemical part it would have been more interesting to discuss the possible tautomerism of1,2,4-triazol-3-thione and 1,3,4-thiadiazol-2-amine. For 1,2,4-triazol-3-thione see : Bipan Dutta et al.,“Prototropic tautomerism of 4-Methyl 1,2,4-Triazole-3-Thione molecule in solvent water medium:DFT and Car–Parrinello molecular dynamics study”, Chemical Physics 463 (2015) 30–37. DOI:10.1016/j.chemphys.2015.09.0175-

Response:This issue was briefly discussed and added to the chemistry part with the insertion of the above mentioned reference for further explanation.

  1. The number 2 in NH2 should be as subscript (NH2) as well as 1H and 13C should be as superscript(1H; 13C).

Response:The above mistyped words was corrected.

  1. At row 187 is reported for Taxol an IC50 =7,8 μM on MCF-7 cell line. I don’t understund why it is veryfar from the value reported in literature on MCF-7 cell line: IC50 = 0.0025μM corresponding to 2.5Nm (J.E. Liebmann et al. “Cytotoxic studies of pacfitaxel (Taxol®) in human tumour cell lines” Br. J. Cancer(1993), 68, 1104-1109. DOI: 10.1038/bjc.1993.488. This value is in the same order of magnitude as confirmed in 2014 in the reports by NCI for the 60 full panel cancer cell lines (8,8nM for MCF-7 and1.95nM for HCT-116 cell line respectivelly).

Response: Maybe that is due to different sensitization and resistance of the used breast cancer cells to taxol. In addtion, taxol concentation and time of exposure may also play a role in that. For example M.N Motiwala et at reporetd IC50 value for 24 h exposue to taxol as 0.22± 0.003 μM in publicaytion Synergy, volume 2, issue 1, 2015, pages 1-6.

  1. The tables should be formatted as in the IJMS template and not like those reported in the manuscript.

Response: The Instructions for Authors were accurately revised and the tables were modified as recommended in the provided template Form.

  1. In Table 2 is very strange that the standard deviation (S.D.) values are so huge in comparison with the inhibition values. The authors have performed a statistical test to verify and validate the obtained results? Have IC50 and SD been put reversed?

Response: The SD values were revised and corrected.

  1. In Table 2 are the reported Log P values experimental or calculated? In this latter case which kind ofLog P was used. They are many algorithms able to calculate Log P and based on different approachesas fragmental , atom-based and conformation dependent. The values obtained from differentmethods are often very different.

Response: it was calculated using MOE software (Labute, P.; MOE LogP(Octanol/Water) Model unpublished. Source code in $MOE/lib/svl/quasar.svl/q_logp.svl (1998).)

  • https://cadaster.eu/sites/cadaster.eu/files/challenge/descr.htm

  1. Paragraph 2.2.7 is only a report of the structure of original ligand of 5YJB and it looks unnecessary, as well as figure 6.

Response: The authors presents this paragraph in addition to figure 6 to explain the original ligand binding which was drawn separately to be compared to our compounds of interest  and to clarify the relationship between it and the tested compounds under the same environment of experimental calculations.

  1. Figures 6-10 reported in the manuscript don’t follow the indications present in the template or in theguide to the authors. In addition the resolution and the dimension ratio are not correct and all figures appear as "stretched".

Response: The resolution of the figures has been adjusted to meet the reviewrs requirements.

  1. Paragraph 2.2.8 regarding conformational analysis is unuseful, because during the docking processall the rotatable bonds can assume any conformation which adapt better to the binding pocket andnot necessarily to the conformation with the lowest energy value.

Response: The authors believe that the conformationl analysis study is important for understanding the stability of different isomers by taking into account the spatial orientation and through-space interactions of substituents . It was performed prior to docking for accurate results.

  1. Figure 7, which is unuseful (as written in the point above), reports the structures of molecules notwith the same atoms with the same color. All this makes interpretation difficult for the reader. Onlyimage in (d) has aproximately the standard colors for the different atoms.

Response: In Figure 7, the atoms of variable compounds were carefully chosen to be different from each other for remarkable discrimination between the tested compounds by their different colors.

  1. Given the speed with which the results of molecular docking are obtained, the authors should have performed the calculations on all the tested molecules.

Response: The molecular modeling calculations were carefully performed to selected derivatives regarding to their biological data to explain the difference between the most and least active candidates.

  1. Paragraphs 2.2.10 and 2.2.11 report only a 3D view of interactions involved between the ligand andthe binding pocket. The figure and the captions are not clear. Which kind of surface map are takeninto account? It is all confused.

Response: Paragraph 2.2.10 reports Contact Statistics application which is used to calculate, coordinates of preferred locations for hydrophobic and hydrophilic ligand interactions meaning that (study the interactions between the chemical components of the ligands and the protein microenvironment surrounding them) where Paragraph 2.2.11 surface map is more related to the protein (not the ligand ) hydrophobicity effects on anticancer activity. Both applications are complementary to complete other to complete the picture regarding explaining the biological activity. 

  1. At row 422 the authors have cited as reference compound for docking calculations GSK. Why theyhave not used also the original ligand (4-[5-(piperidin-4-ylmethoxy)-2-(p-tolyl)pyridin-3-yl]benzonitrile) present in the binding pocket of LSD1-CoREST (5YJB)?

Response: The authors have cited as reference compound for docking calculations GSK and not co-crystallized compound due to the following reason: GSK-LSD1 dihydrochloride is a potent, selective and irreversible lysine specific demethylase 1 (LSD1) inhibitor with an IC50 of 16 nM due to its strong activity.

  1. It is necessary to add a summary table reporting the energy values of docking and the types of interactions between each small molecule and the amino acidic residues involved in the interactions.

Response: The energy calculations and aminoacid residues were summarized in table 7 as requested 

  1. In “Concluding remarks“ the authors report :” Most of the synthesized derivatives had potency withIC50 at submicromolar and selectivity performance comparable to reference inhibitors”, but it is notpossible to talk about of “potency” in enzyme inhibition, as well as of “selectivity” with respect towhat? It was used only a single type of enzyme.

Response: Thanks for you, we did potency for LSD1 and compared results with highly similar target MAO, so we referred here for potency against one enzyme and selectivity over MAO target.

  1. In the experimental part are missing the color and the physical state of compounds. Are they crystalline or amorphous powder?

Response: They are all crystalline powders with yellowish to white colors and are described in the first part of the chemistry discussion.

  1. The melting point for compound 19 (row557) results as in a large range of values (10 Celsius degrees),why? Is it impure?

Response: Thanks for you, No it was very pure but the melting was wrong and corrected. 266-268.

  1. TLC data are missing, as well as the eluents compositions and Rf value of each compound.

Response: This point has been addressed in the revised version.

  1. Which is the purity of compounds used in biological assay? How was it determined?

Response: The sample’s purity for biological data is defined based upon multiple recrystallization processes and the analysis of spectral data and images of TLC workusing UV fluorescent Silica gel Merck 60 F254 plates, and the spots were visualized using a UV lamp (254 nm). In addition, the elemental analysis gave the purity show of each compound from CHNOS %.

  1. Although there is no obligation to report mass spectra in low and high resolution, it is desirable that in the year 2022, they are essential data to insert in the characterization of a compound, instead touse the elemental analysis, which can be easily invented.

Response: Thanks for paying attention to this point. We will consider it in our future investigation for sure.

  1. In paragraph 4.2: how many independent experiment were done?

Response: Three as stated down in legend of Table1.

  1. Paragragh 4.9 Molecular Docking needs to be added of more specific calculations details andinformation on protein structure preparation. The minimization of protein (5YBJ) was obtained withor without original ligand (8WC) and FAD?

Response: The required information was adjusted and Paragragh 4.9 was modified to meet the reviewer’s comments.  The minimization of protein (5YBJ) was obtained without the original ligand (8WC)

  1. The authors report : “The molecular mechanics force field 'AMBER' was used to do energyminimization”, but they are many types of AMBER force field (AMBER94, AMBER99, AMBER10:EHT,AMBER12:EHT…and so on), in addition not all the AMBER ff are parametrized for small moleculesbut only AMBER10:EHT, AMBER12:EHT and following.

Response:The molecular mechanics force field 'AMBER' was used to do energy minimization”, using  AMBER force field as implemented in MOE program to give the best results.

  1. At row 673 it is reported: “…minimizations (EM) were carried out for each benzimidazolederivative…” but no benzimidazolic structure is present in compounds 18-28.

Response: The word benzimidazoles was omitted  and replaced by (each selected candidate) as requested.

  1. The authors at row 675 wrote: “A radius of 10.0 Å was used to detect the enzyme's active site”, butthe active site of the enzyme is known, due to the presence of ligand (8WC) in the X-ray structure .

Response: A radius of 10.0 Å was used to detect the enzyme's active site as we remove the ligand prior to docking to precisely determine the docking area in the receptor and to easily validate the results.

  1. All the paragraph 4.9 is very confused both as calculation information and the order of information.

Response: The required information was adjusted and Paragragh 4.9 was modified to meet the reviewers comments.

- In conclusion to be considered for the publication in the important journal as International Journal of Molecular Sciences (Journal Rankings Q1, impact factor 5.923), all the manuscript needs to be rewrittenand improved in many parts as well as it needs to be added of further experiments (mass spectra in lowand high resolution, molecular docking on all the series of tested compounds and ADME prediction (freeavailable by the web server SwissADME: http://www.swissadme.ch/) in order to evaluate the potentialbioavalaibility of the most promising compounds. I would like to add that, although it is possible tovalidate the interactions between a small molecule and a target protein after have performed thebiological assays, when a project is based on the molecular design of potential bioactive compounds,molecular docking, molecular dynamics and Drug-Likeness ADME prediction should be obtained beforeto synthesize the molecules, in order to select the best candidates.In any case I firmly believe that the manuscript, written in this manner, doesn’t fit exactly the SpecialIssue: “Malignant Melanoma: Molecular Mechanism and New Agents for Prevention and Therapy”because all the study here reported is focused on breast-cancer (MCF-7), colon cancer (HTC116) andliver cancer (Hep-G2) cell lines. Only some generic keywords reported in the list of Special IssueInformation are present in this manuscript, but I think that these keywords need to be associated to theword melanoma, otherwise every type of cancer could be considered, making the special issue no longersuch.Both the articles already published in this special issue are centered on melanoma.I address the authors to publish this manuscript elsewhere.

Responses to final major conclusion points:-

  1. Regarding to mass spectra for novel compounds.
  • Thanks for your kind advice, but we did NMR and elemental analysis with TLC and multiple recrystallization work. I think it is sufficient for compound verification.
  1. Docking all compounds prior to synthesis and biology
  • For our group, we are professional in design of targeted triazole compounds based upon our experience and we proceed from medicinal chemistry background to synthesis and biology, further we try to map SAR analysis through different computational methods like docking or surface map analysis for compound optimization. So needless to start with in silico experiments to suggest which compound will be active or not.
  1. Calculation of ADME properties for compounds
  • Thanks a lot, we inserted table under pharmacokinetic section for most active compared to less one and reference drug.
  1. Relationship of work with special issue
  • Thanks for you, here we tried to show the direct relationship of LSD1 to major types of cancer and we proved that which indirectly is highly related to other types of cancers like melanoma. So we send the manuscript to journal special issue.

Round 2

Reviewer 2 Report

Please, see attached file

Author Response

Reviewer 1:

This reviewer would like to thank the authors for the effort carried out for enhancing the quality of the manuscript. However, some important points still need to be improved before considering the paper for publication:

  • The English grammar still need improvement, mainly in the abstract.

Response: Thanks for you, English grammar have been checked and revised well especially abstract.

  • The chemical structure of the compound described were included in the NMRs, however no iconology was represented to relate the signals in the spectrum with their corresponding H or C in the chemical structure. Doing this would make the understanding of the spectra much easier for the reader.

Response: This comment has been addressed in the revised version of the supplementary.

Author Response

Reviewer 2:

  1. Figure 6, 8, 10, 11 needs to be redone following the “instruction for authors” of Int. J. Mol. Sci.
  2. Improve the resolution

Response:  The resolution of figures was improved to meet journal requirement.

  1. remove the red frame

Response:  Removed

  1. correct the ratio between height and length, to eliminate the "stretching" effect, present in all the figures shown.

Response: It was modified and kept without stretching.

  1. Remove paragraph 2.2.8 and figure 7, because are unnecessary, as they do not provide useful information.

Response: Removed and figures reordered.

  1. Remove paragraph 2.2.10 and fig. 9 for the same reasons reported above.

Response:  Removed and figures reordered.

  1. In experimental part, instead to use “pellets” use powder.

Response: As requested by the reviewer, completed.

  1. Check the references, because some of them are incomplete.

Response: Checked and were completed.